

# Deep learning-based recognition system for pashto handwritten text: benchmark on PHTI

Ibrar Hussain[1,2], Riaz Ahmad[1], Khalil Ullah[3], Siraj Muhammad[1], Rasha Elhassan[4] and Ikram Syed[5]

[1] Department of Computer Science, Shaheed Benazir Bhutto University, Sheringel, Dir, Pakistan
[2] Department of Computer Science & IT, University of Malakand, Chakdara, Pakistan
[3] Department of Software Engineering, University of Malakand, Chakadara, Pakistan
[4] Department of Computer Science, King Khalid University, Abha, Saudi Arabia
[5] AI and Software, Gachon University, Seongnam-si, Republic of South Korea

## ABSTRACT

This article introduces a recognition system for handwritten text in the Pashto language, representing the first attempt to establish a baseline system using the Pashto Handwritten Text Imagebase (PHTI) dataset. Initially, the PHTI dataset underwent pre-processed to eliminate unwanted characters, subsequently, the dataset was divided into training 70%, validation 15%, and test sets 15%. The proposed recognition system is based on multi-dimensional long short-term memory (MD-LSTM) networks. A comprehensive empirical analysis was conducted to determine the optimal parameters for the proposed MD-LSTM architecture; Counter experiments were used to evaluate the performance of the proposed system comparing with the state-of-the-art models on the PHTI dataset. The novelty of our proposed model, compared to other state of the art models, lies in its hidden layer size (*i.e.*, 10, 20, 80) and its *Tanh* layer size (*i.e.*, 20, 40). The system achieves a Character Error Rate (CER) of 20.77% as a baseline on the test set. The top 20 confusions are reported to check the performance and limitations of the proposed model. The results highlight complications and future perspective of the Pashto language towards the digital transition.

## INTRODUCTION

One of the essential themes of the digital turn (*Rosenzweig, 2003*) is that media stored in computers must be adaptable for analysis and effective processing in various applications. Document images are media contents that requires easy analysis and processing. These images include handwritten documents, forms, bank cheques, reports, printed magazines, historical documents, *etc*. Cameras and scanners serve as the primary sources for the acquisition of document images. However, these images (documents) are initially in pixel form and cannot be explicitly analyzed and processed. Therefore, a special mechanism is needed to convert such document images into a suitable form that can be read, analyzed

Corresponding authors
Rasha Elhassan,
relhassan@kku.edu.sa
Ikram Syed, ikram@gachon.ac.kr

and processed for applications including Natural Language Processing (NLP), text recognition, speech recognition and translation services.

Document image analysis (DIA) is a research area that develops a system to obtain computer-readable text from document images. The output can be used to recognize, store, retrieve and manipulate for other applications (*Ahmad & Fink, 2019*). The DIA's main component is optical character recognition (OCR) (*Kasturi, O'gorman & Govindaraju, 2002*), often referred to as text recognition and has been considered an area of pattern recognition. Languages like English, Dutch, French, Chinese, Japanese, Arabic, *etc.*, have plenty of work regarding OCR (*Memon et al., 2020*).

There are effective and commercially available OCR systems like Nanonets, ReadIRIS, ABBYY FineReader, Kofax OmniPage, Adobe Acrobat ProDC, Tesseract, and SimpleOCR (https://www.adamenfroy.com/best-ocr-software). Although these OCR systems can recognise printed and handwritten text from several renowned languages, the Pashto language still needs comprehensive work to reach a maturity level in OCR systems. Further, the Pashto handwritten text is thought to be more complicated than printed text, and there is little work that addressed the recognition of the Pashto handwritten materials (*MacRostie et al., 2004*; *Wahab, Amin & Ahmed, 2009*; *Ahmad et al., 2015a*, *2015b*, *2016*, *2017*; *Ahmad, Naz & Razzak, 2021*; *Khan et al., 2021*, *2020*, *2019*; *Amin, Yasir & Ahn, 2020*; *Uddin et al., 2021*; *Rehman et al., 2021*; *Jehangir et al., 2021*).

The relevant research that addresses the Pashto language in the field of DIA can be divided in two field; one focused on the recognition of isolated characters and digits, and the other is focused on the recognition of text-lines. The later work has the ability to be generalized in terms of utilization. However, only two works can be found that address the recognition of the Pashto text lines, *i.e.*, *MacRostie et al. (2004)*, *Ahmad et al. (2016)*. Looking in depth at these two works (*MacRostie et al., 2004*; *Ahmad et al., 2016*), we found that the BBNBYBLOS work was based on Hidden Markov Model (HMM) (*Rabiner & Juang, 1986*), and the dataset is not publically available for benchmarking the latest technologies. On the other hand, the work presented in *Ahmad et al. (2016)* clearly limits the material that is written by Katibs (calligraphers). Such material has a very close resemblance with printed text and is considered less hard compared to handwritten text. As a result, there is a clear gap that is present regarding the materials of handwritten Pashto text. Particularly, the real-world text-lines presenting the Pashto language with handwritten samples have not been explored so far.

This article recognizes Pashto handwritten text using MD-LSTM (*Graves & Schmidhuber, 2008*) architecture. Several experiments figure out the best hidden layer size for the proposed MD-LSTM system as a benchmark on the PHTI dataset (*Hussain et al., 2022*). The proposed architecture is different from the model of *Graves & Schmidhuber (2008)* and from the model of *Ahmad et al. (2016)* in terms of number of hidden layer size and *Tanh* layer size for LSTM units. Our architecture consist of three hidden layers containing $(10, 20, 80)$ LSTM units respectively and $(20, 40)$ *Tanh* layer size, while the Alex Graves's model contains three layers containing $(2, 10, 50)$ LSTM units and *Tanh* layer size is $(6, 20)$ (*Graves & Schmidhuber, 2008*).

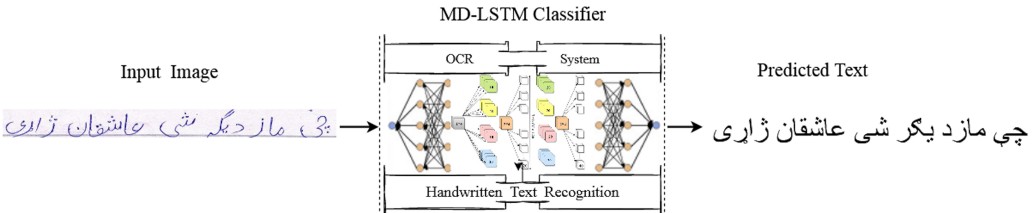

**Figure 1** Framework for Pashto handwritten text recognition system.

Similarly, the model of *Ahmad et al. (2016)* contains three layers of $(4, 20, 100)$ LSTM units with *Tanh* layer size of $(16, 80)$. From the available literature, this is the first time to evaluate the PHTI dataset for real world text-lines of the Pashto language. Figure 1 illustrates the abstract view of our proposed system.

The rest of the article is organized as follows: "Related Work" presents the DIA's Pashto work. "Pashto Handwritten Text Imagebase-PHTI" describes the PHTI dataset. The proposed methodology is described in "Proposed Recognition System for PHTI". "Experiments and Results" covers the experimental setup, while "Conclusion" covers the conclusion.

## RELATED WORK

In general, there is enough work available on the Pashto language. However, regarding the recognition of handwritten Pashto text, there is relatively little work. Therefore, we present relevent work that may help in the recognition of handwritten Pashto text. The following texts describe several techniques based on different approaches, including principal component analysis (PCA), hidden Markov model (HMM), artificial neural networks (ANN), convolutional neural networks (CNN), K-nearest neighbor (KNN), recurrent neural network (RNN), histogram of gradients (HoG), zoning features, Scale-Invariant Feature Transform (SIFT) *etc.*

*MacRostie et al. (2004)* presented a DIA system for the Pashto language with the name of BBN Byblos. The BBN system is based on HMM and their CER ranges from 2.1% to 26.3%. However, the dataset they used in their research is not available. A dataset of 1,000 syntactically generated images of Pashto ligatures was presented by *Wahab, Amin & Ahmed (2009)*. They have used PCA for ligature classification and obtained 17% accuracy as average. Due to the scale variations in the dataset reported in *Wahab, Amin & Ahmed (2009)*, *Ahmad, Amin & Khan (2010)* tested SIFT descriptor for classification of those ligatures and achieved 73% accuracy and performed better than PCA.

Another work, *Ahmad et al. (2015a)*, used MD-LSTM showing promising results by recognizing ligatures with scale and rotation variations. The system achieved a ligature recognition rate of 99%. A dataset named Katib's Pashto Text Imagebase (KPTI) was introduced by *Ahmad et al. (2016)* in 2016. The KPTI dataset contains 1,026 images taken from Pashto-scribed books. The obtained images have been further split into 17,015 text-line images using a methods described in *Ahmad et al. (2017)*, *Ahmad, Naz & Razzak (2021)*. The CER on the KPTI dataset was 9.22% using MD-LSTM model. The KPTI

dataset was further evaluated in *Ahmad et al. (2018)*, where the issue of space anomaly was considered, and hence the accuracy was improved by 2.89%. Their approach for handling space anomaly was also validated on Arabic as well as on Urdu text. So far, the related work presented here contains research on real world Pashto text lines. The rest of the work only addressed either individual characters or digits in the Pashto language. Such work is not considered to be fit for generalization to explore language composition and structure.

A medium-sized dataset was presented by *Khan et al. (2019)* containing 102 samples of 44 Pashto characters produced a total of 4,488 images in the dataset. They used both KNN and ANN and obtained an accuracy of 70.05% and 72% respectively (*Khan et al., 2020*) presented a Pashto Handwritten Numerals Database (PHND) containing 50,000 scanned Pashto images for digits. They used CNN and RNN models to evaluate their dataset and obtained an accuracy of 98%.

Likewise, another medium-sized dataset was created by *Khan et al. (2021)* for the Pashto handwritten characters recognition. There are 8,800 images of Pashto characters in the dataset. They used techniques like zoning techniques, Gabor filters and hybrid feature maps. Hybrid feature maps showed a promising result by obtaining an accuracy of 83%. *Amin, Yasir & Ahn (2020)* in 2020, created a dataset named Poha containing 26,400 images of 44 Pashto characters and 10 Pashto digits.They used CNN-based model to evaluate the Poha dataset and has obtained an accuracy of 99.64%.

Similarly, *Uddin et al. (2021)* presented the Pashto handwritten character dataset, consisting of 43,000 images. The authors used a deep neural network with a ReLU activation function and obtained an accuracy of 87.6%. Another medium-sized dataset was also created by *Huang et al. (2021)*. Their dataset contains 11,352 images of Pashto characters. They used KNN algorithm with HoG and zoning features and achieved an accuracy of 80.34% on KNN and 76.42% on HoG.

*Rehman et al. (2021)* developed a dataset for Pashto digits and characters. The authors used LeNet, CNN and Deep CNN models for Pashto digits and character recognition. In their work, the Deep CNN shows promising result and yielded an accuracy of 99.42% for handwritten Pashto digits and 99.17% for handwritten characters.

*Siddiqu et al. (2023)* developed an isolated Pashto character database for 44 Pashto alphabets, using a variety of font styles. The database was pre-processed and reduced in size to 32 by 32 pixels, followed by conversion to binary format with a black background and white text.

*Shabir et al. (2023)* prepared an extensive Pashto-transformed invariant inverted handwritten text dataset. They fine tuned the MobileNetV2 for Pashto (text classification and extracting images features). The (TILPDeep) transformed invariant lightweight Pashto deep learning techniques was used and the authors achieved an accuracy of 0.9839 on training set, and 0.9405 for the validation set.

*Khaliq et al. (2023)* work for the recognition of Pashto characters and ligatures of handwritten text, the PHWD-V2 dataset was used for the experiment. Several machine and deep methods were assessed, including MobileNetV2, VGG19, MobileNetV3, and customized-CNN (CCNN). The CCNN shows maximum accuracy on 93.98, 92.08, and 92.99 for training, validation, and testing, respectively.

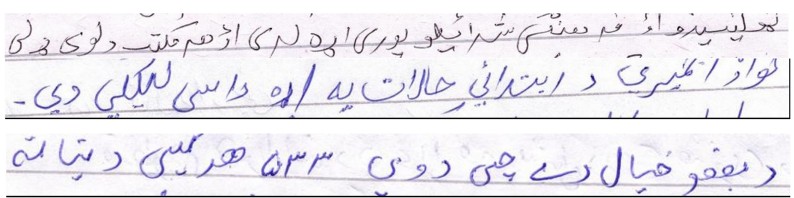

**Figure 2  Samples of the PHTI dataset.**               

*Hussain et al. (2022)* published a dataset named Pashto Handwritten Text Imagebase (PHTI). The PHTI contains 36,082 text line images presenting real world Pashto text in handwritten form. However, the dataset is not explored or tested for character recognition.

Hence, from the available literature, very little work addresses handwritten text-lines from real world data recognition for Pashto. Most of the work addressed isolated characters or digits. The main reason was the non-availability of the dataset that contains images of handwritten text lines. Therefore, this work presents a baseline recognition system on PHTI dataset.

## PASHTO HANDWRITTEN TEXT IMAGEBASE-PHTI

In this research, we use PHTI (*Hussain et al., 2022*) dataset as a benchmark. The dataset is designed for research by considering the generalization aspect of the Pashto language. PHTI contains 36,082 Pashto handwritten text-line images along with ground truth annotated with UTF-8 codecs. PHTI comprises various genres, including short stories, historical memory, poetry, and religious content. The data was collected from diverse learners with gender identities, educational backgrounds, and personal experiences. The available version of PHTI has 169, unique characters. By observing the context of Pashto language, some characters are not directly linked. Therefore, this study suggests pre-processing. Figure 2 shows samples of PHTI text-line images. The dataset can be downloaded from https://github.com/adqecsbbu/PHTI.

### Pre-processing

As mentioned, the PHTI has 36,082 text-line images. We discovered non-Pashto/unwanted characters such as A-Z, a-z, 0-9 and special characters such as (¡, @,!). The text-line images with these unwanted characters are skipped. The number of text-line images decreases from 36,082 to 25,939. Unique classes/letters are reduced from 169 to 94. Furthermore, the variable height of the text-line images have been fixed to 48 pixels by locking the aspect ratio to reduce the training time.

### Data split

To support supervised learning, the data must be split fairly into training, validation and test sets. For this, we split the PHTI data using the holdout method (*May, Maier & Dandy, 2010*). The data was shuffled, and then 70% of text lines were selected for the training, 15% for validation and 15% for the test.

## Evaluation metric

To evaluate the results of our experiments on the PHTI dataset, we used the Levenshtein Edit Distance (LED) (*Yujian & Bo, 2007*). The LED is a metric that measures the difference between two sequences given as strings. The overall error is determined by calculating the total number of insertions (I), substitutions (S), and deletions (D) divided by the total number of characters in the target string (N). Equation (1) shows how CER can be computed using LED.

$$CER = \frac{(I + S + D)}{N} \tag{1}$$

# PROPOSED RECOGNITION SYSTEM FOR PHTI

Handwritten text recognition is a classic issue; because of language specificity, the solutions are always diverse and based on language-specific treatments. However, RNNs are the most suitable among other solutions. In such problems, RNN-based techniques have performed better than other approaches (*Naz et al., 2017*; *Messina & Louradour, 2015*). The proposed model and grid search for the optimal parameters is essential before going into recognition system detail.

## Proposed model

Deep-learning models based on RNNs can better learn the previous context of input sequences. Classic RNNs, however, are limited by the vanishing gradient problem, especially for long-term dependencies (*Hochreiter, 1998*). LSTM was introduced by *Hochreiter & Schmidhuber (1997)* to address the vanishing gradient problem; LSTMs learn both short-term and long-term dependencies. We use multi-dimensional LSTM (MD-LSTM) based RNN approach that can scan the input images in four directions (Up and Down, Left and Right). The MD-LSTM is robust against many variations include scale, rotation, and registration.

The proposed MD-LSTM system has two major components, (1) the hidden layer size and (2) *Tanh* layer size. The optimal values for hidden layer size and *Tanh* layer size are considered to be crucial. For example the MD-LSTM model proposed by *Graves & Schmidhuber (2008)* contains three hidden layers with 10, 20, 80 LSTM units while another model introduced by *Ahmad et al. (2016)* contains three layers of 4, 20, 100 LSTM units. Therefore, the number of hidden layer size and LSTM units depends on the problem specificity. We need empirical analysis to find the optimal hidden layers as well as the LSTM units in those layers.

In addition to that we also need optimal values for *Tanh* layer size. Therefore, we performed two different grid analysis, (1) for finding the optimal hidden layer size with appropriate LSTM units and (2) for finding the optimal *Tanh* layer size. The next section describes the detail about the overall grid analysis.

**Table 1  MD-LSTM empirical analysis: hidden layers *vs Tanh* layer size where HLS represents Hidden Layer size, TLS represents *Tanh* Layer size, *VS* represents validation set, TS represents test set, TE represents total epoch, PETS represents per-epoch time spent, respectively.**

| S.No | HLS (MD-LSTM) | TLS | VS (CER%) | TS (CER%) | TE | PETS HH:MM:SS |
|------|---------------|---------|-----------|-----------|-----|----------------|
| 1 | 4, 15, 80 | 16, 60 | 23.35 | 23.38 | 62 | 01:05:07 |
| 2 | 5, 20, 75 | 20, 80 | 23.00 | 22.73 | 76 | 01:17:27 |
| 3 | 5, 25, 75 | 20, 100 | 22.45 | 22.31 | 69 | 01:30:45 |
| 4 | 5, 25, 100 | 20, 100 | 22.64 | 22.67 | 58 | 01:48:51 |
| 5 | 6, 15, 80 | 24, 60 | 22.32 | 22.32 | 95 | 01:15:34 |
| 6 | 6, 25, 80 | 24, 100 | 22.51 | 22.57 | 57 | 01:38:22 |
| 7 | 8, 15, 80 | 32, 60 | 21.45 | 21.33 | 84 | 01:28:16 |
| 8 | 8, 25, 100 | 32, 100 | 21.99 | 21.85 | 70 | 01:59:54 |
| 9 | **10, 20, 80** | **40, 80** | **21.58** | **21.09** | **54** | **02:47:24** |
| 10 | 10, 25, 80 | 40, 100 | 21.64 | 21:43 | 57 | 03:15:25 |

**Note:**
   Optimum values are shown in bold.

## Grid analysis

The purpose of the first grid analysis is to obtain optimal size of hidden layer size with LSTM units. For this purpose, we carried out 10 experiments. We started with minimum LSTM units with three Hidden Layers and the focus was on to get minimum CER on test set (TS) and validation set (*VS*). Table 1 shows the overall grid analysis. It is noteworthy that *Tanh* layer size for each hidden layer configuration was taken as at most. For example, consider the first row in Table 1 for hidden layer (*i.e.*, 4, 15, 80) the first *Tanh* layer (*i.e.*, 16) will be in between first and second hidden layer and the second *Tanh* layer (*i.e.*, 60) will be in between second and third hidden layer. Now 4 LSTM units in four different directions (Up, Down, Left, and Right) make 16 connections. Therefore, the maximum *Tanh* layer size is taken as 16. Similarly, for second hidden layer that contains 15 LSTM units and multiplied by four directions makes it 60 connections. Thus the second *Tanh* layer size is taken as 60. After the completion of 10 experiments, we found the optimal values for hidden layer size of (10, 20, 80) LSTM units as shown in bold in Table 1 on serial no 9.

The second grid analysis is made for finding the optimal values for *Tanh* layer units. Therefore, we have performed seven experiments by keeping the hidden layer size of (10, 20, 80) LSTM units as fixed. Table 2 shows the grid analysis for finding the optimal *Tanh* layer size. In this analysis, the focus was also on the minimum CER on TS as well as *VS*. Initially, the size for *Tanh* layer was kept minimum and was gradually increased. In this way, an optimal values of 20, 40 for *Tanh* layer size has provided better performance.

Consequently, the configuration of the proposed model is completed. Now, it has three hidden layers with 10, 20, 80 LSTM units that can scan an input image in four directions. Further more, the *Tanh* layer size of proposed model is 20, 40. Figure 3 shows the detail architecture of our proposed MD-LSTM model.

**Table 2 Grid analysis for *Tanh* layer size where TLS represents *Tanh* layer size, *VS* represents validation set, TS represents test set, TE represents total epoch, PETS represents per-epoch time spent, respectively.**

| S.No | TLS | VS (CER%) | TS (CER%) | TE | PETS HH:MM:SS |
|---|---|---|---|---|---|
| 1 | 05, 10 | 33.34 | 32.98 | 106 | 02:02:47 |
| 2 | 10, 20 | 21.80 | 21.28 | 136 | 02:08:48 |
| 3 | 15, 30 | 22.56 | 22.20 | 102 | 02:14:11 |
| 4 | **20, 40** | **21.15** | **20.77** | **82** | **02:21:31** |
| 5 | 20, 60 | 22.28 | 22.03 | 96 | 02:28:11 |
| 6 | 30, 50 | 22.70 | 22.25 | 83 | 02:20:28 |
| 7 | 30, 70 | 22.49 | 22.15 | 79 | 02:38:23 |

Note:
Optimum values are shown in bold.

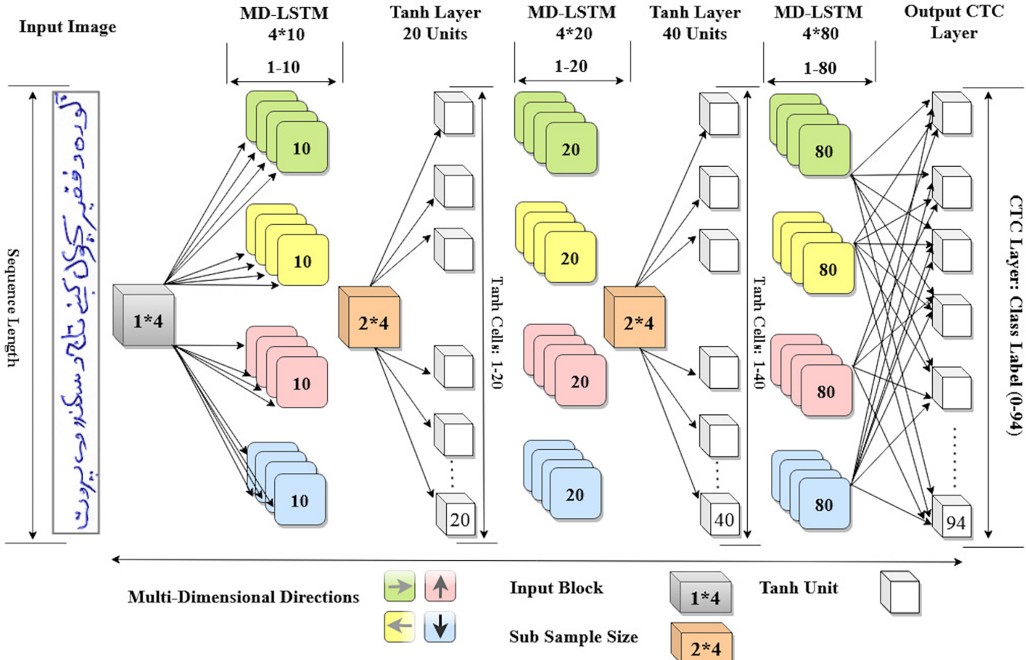

**Figure 3 Proposed MD-LSTM model consists of three hidden layers with LSTM units and two *Tanh* layers with *Tanh* activation function.**

# EXPERIMENTS AND RESULTS

After grid search, finding the suitable hidden layer size (*i.e.*, 10, 20, and 80) containing LSTM units, and *Tanh* layer size (*i.e.*, 20, 40) for the PHTI dataset. The proposed model was trained on the train set. To avoid any over-fitting, the validation set was provided during the training process to check the validity of the learning process. After 70 epochs, the training converged to minimal loss and has shown CER of 15.45% on the training set, while 21.15% on the validation set. Then the training was stopped and the model was checked for unseen data *i.e.*, test set. As PHTI dataset is not tested so far, therefore, no results available for comparison. However, to validate the selection of our proposed model,

**Table 3 Comparison of *Graves & Schmidhuber (2008)*, *Ahmad et al. (2016)* and proposed models.**

| Model | Training % | Validation % | Testing % |
| --- | --- | --- | --- |
| Proposed | 82.99 | 79.13 | 79.23 |
| *Graves & Schmidhuber (2008)* | 70.00 | 68.83 | 69.05 |
| *Ahmad et al. (2016)* | 81.87 | 76.00 | 76.06 |

**Table 4 Top 20 confusions, associated with the letters in the PHTI dataset where MCWL represents misclassifications with letter, and MCWAL represents misclassifications with all letter, respectively.**

| S.No | Target Letter | Top MCWL | MC WAL | S.No | Target Letter | Top MCWL | MC WAL |
| --- | --- | --- | --- | --- | --- | --- | --- |
| 1 | ي | 869 | ى 2,063 | 11 | ب | 164 | پ 608 |
| 2 | ى | 833 | ي 1,803 | 12 | ل | 87 | ک 557 |
| 3 | ر | 411 | د 993 | 13 | پ | 172 | ب 529 |
| 4 | ن | 258 | ت 921 | 14 | ة | 426 | ه 503 |
| 5 | ه | 127 | ة 892 | 15 | ى | 139 | ي 462 |
| 6 | د | 295 | ر 882 | 16 | ا | 60 | ل 426 |
| 7 | م | 83 | ه 852 | 17 | خ | 63 | ح 399 |
| 8 | و | 288 | د 820 | 18 | ع | 58 | م 371 |
| 9 | ت | 262 | ن 642 | 19 | ښ | 46 | م 353 |
| 10 | ک | 191 | ك 626 | 20 | ف | 54 | خ 327 |

we have done two extra experiments (1) for examining the models of *Graves & Schmidhuber (2008)* and (2) *Ahmad et al. (2016)* on PHTI dataset. The results show that our proposed model outperforms *Graves & Schmidhuber'a (2008)* and *Ahmad's et al. (2016)* model by achieving an accuracy of 79.23% on the test set of PHTI dataset. Table 3 shows the comparison of the mentioned models on the PHTI dataset. Further, in terms of complexities present in the material, we computed the overall confusion matrix that is $94 \times 94$ matrix, and for quick reference, Table 4 provides the top 20 confusions associated with the letters in the PHTI dataset. Among these top 20 confusions, the top two confusions are related to the letters and. Both letters are 1,712 times miss classified with each other. The major reason is the shape similarity present among the Pashto letters.

Similarly, the overall confusions of these two letters with all other letters are 3,866 times. Thus the impact of the top two confusions in the overall error is about 2.59%. These findings also validate other research including (*Ahmad et al., 2016*). Visual analysis shows images that present shorter text-lines are comparatively well recognized. However, longer text-lines are not recognized accurately. Figure 4 shows the visual comparison of a few text-line images along with the predicted text. The results also signify the empirical analysis *via* grid search for finding the hidden layer size and *Tanh* layer size, as the proposed model provides better performance compared to *Graves & Schmidhuber*'s *(2008)* and *Ahmad et al.*'s *(2016)* models.

| | |
|---|---|
| Input Image | سخری کوري ا وښتر اونو بمین ا و لو بہ تري نهغان ژ کورلو بیا څہ کورہ |
| Predicted Text | ^سخري کو لی^ د م ۱۸۸ نو ا و ^ پین^ ا و بو بہ تري هغ^ان ژمو ر لو بیا څہ کو رہ |
| Input Image | حیپرمنا رلے چی پہ څہ وخت کنې ہغہ را مح سو سبزي |
| Predicted Text | خ^ ر ست د ے چی پہ څہ و خ^ کی^ هغہ را مح ^و سبرِ ي |

Red= Deletion, Green=Substitution, Blue=Insertion

**Figure 4 Visual comparison of input images with the predicted output.**

## CONCLUSION

This article presented a deep learning-based recognition system using the MD-LSTM model for the recognition of Pashto handwritten text-line images. The PHTI dataset was bench-marked for the first time in this research. Further, for the proposed recognition system, comprehensive experiments were conducted to find the optimal hidden layer size for MD-LSTM-based architecture. Similarly, the *Tanh* layer size was also determined by several experiments. The final proposed model was also compared with the models developed by *Graves & Schmidhuber (2008)* and *Ahmad et al. (2016)*. The proposed system shows a baseline accuracy of 79.23% on the test set of the PHTI dataset. Further, the misclassification is examined, and the top 20 confusions on the PHTI dataset are provided. The majority of the misclassifications are among the letters which have a resemblance in shape. We suggest transfer learning in the near future to improve the overall accuracy of the PHTI dataset.

### Funding

This work was supported by the King Khalid University Deanship of Scientific Research through the General Research Project under grant number (GRP/172/44/1444). The funders had no role in study design, data collection and analysis, decision to publish, or preparation of the manuscript.

### Grant Disclosures

The following grant information was disclosed by the authors:
King Khalid University Deanship of Scientific Research through the General Research Project under grant number: GRP/172/44/1444.

### Competing Interests

The authors declare that they have no competing interests.

### Author Contributions

- Ibrar Hussain conceived and designed the experiments, performed the experiments, analyzed the data, performed the computation work, prepared figures and/or tables, authored or reviewed drafts of the article, and approved the final draft.

- Riaz Ahmad conceived and designed the experiments, performed the experiments, analyzed the data, performed the computation work, prepared figures and/or tables, authored or reviewed drafts of the article, and approved the final draft.
- Khalil Ullah conceived and designed the experiments, performed the experiments, analyzed the data, performed the computation work, prepared figures and/or tables, authored or reviewed drafts of the article, and approved the final draft.
- Siraj Muhammad conceived and designed the experiments, performed the experiments, analyzed the data, performed the computation work, prepared figures and/or tables, authored or reviewed drafts of the article, and approved the final draft.
- Rasha Elhassan conceived and designed the experiments, performed the experiments, analyzed the data, performed the computation work, prepared figures and/or tables, authored or reviewed drafts of the article, and approved the final draft.
- Ikram Syed conceived and designed the experiments, performed the experiments, analyzed the data, performed the computation work, prepared figures and/or tables, authored or reviewed drafts of the article, and approved the final draft.

## Data Availability

The data is available at Zenodo: Ibrar Hussain. (2024). adqecsbbu/Deep-Learning-Based-Recognition-System-for-Pashto-Handwritten-Text-Benchmark-on-PHTI: Benchmark_PHTI (v0.1.0). Zenodo. https://doi.org/10.5281/zenodo.10527848.

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
