# Peer review of "Deep learning-based recognition system for pashto handwritten text: benchmark on PHTI"

_PeerJ Computer Science, doi:10.7717/peerj-cs.1925_

## Round 0.1 · original submission · Major Revisions

Based on the reviewers comments I recommend to update the paper revision.

**Language Note:** The review process has identified that the English language must be improved. PeerJ can provide language editing services - please contact us at copyediting@peerj.com for pricing (be sure to provide your manuscript number and title). Alternatively, you should make your own arrangements to improve the language quality and provide details in your response letter. – PeerJ Staff

Reviewer 1 ·

Basic reporting

1. The English language needs improvement.
2. Sufficient background on the topic is given with enough literature review.
3. Figures and table are presented in a manner that are readable and concise.
4. Results are presented that validate the effectiveness of the proposed methodology.

Experimental design

1. The paper falls well under the scope of the journal.
2. Problem statement is given such that the research questions are well highlighted and explained.

Validity of the findings

1. The findings are thoroughly discussed hence highlighting the effectiveness of the method for the task at hand.
2. However, the authors should elaborate more on the the limitation and future work.

Additional comments

Following are the additional comments:
This manuscript presents a Deep learning-based recognition system for Pashto handwritten text. The proposed model shown to achieve high accuracy compared to the other models. But I do have several grievances and suggestions that should be addressed before considering the publication of this manuscript.
1. The research gap described in the introduction seems not to be sufficient; also the introduction is a bit short.
2. Please height the main contribution of this work which could be easy for a reader to identify the authors contribution of this work to the Pashto language recognition.
3. Why BLSTM & CNN are not used.
4. What are the impacts of this work for Pashto?
5. The choice of references should be carefully checked and the right citations should be audit in this manuscript.
6. What are the scopes of this work in term of other cursive script languages?
7. What are the limitation and future work?
8. Can this work be link with academia and industry?

Reviewer 2 ·

Basic reporting

Overall, I find the manuscript to be well-prepared and suitable for publication in this journal. But, I would like to raise a few important points for the author's consideration.

Experimental design

1. I would recommend rewriting the Introduction section and highlight the key aspects of the research.
2. The related work section is not well-structured. Classify the work in meaningful categories and improve it with more recent and related works.
3. It would be great to highlight some description about the weaknesses of the proposed approach.
4. The figures should be enlarged for better visibility.
5. The captions of the figures and tables should be extended for better understanding. For example, In Fig 5, Fig 6, and Fig 7, only references are provided in the captions.
6. Why the accuracy metric and LED metric are only used for performance analysis?

Validity of the findings

no comments

Reviewer 3 ·

Basic reporting

The manuscript focuses on a deep learning-based recognition system for Pashto handwritten text. The author explores the application of MD-LSTM and evaluates the results based on widely used performance criteria. Overall, I find the manuscript to be well-prepared.

Experimental design

The experiments are appropriately designed.

Validity of the findings

All the finding looks valid and clearly understandable.

Additional comments

I have a few suggestions for further improvement.

1. The research article (KPTI: Katib Pashto Text Image base and deep learning benchmark) has already developed the database with the same idea in the same language, then what is the significance of this research work? Clearly highlight your contribution in this work.
2. How was the dataset split for training, test and validation?
3. Is it possible to test the proposed scheme on real images from mobile camera?
4. Can it be generalized on the Persian, Urdu and Arabic handwritten text?
5. Why was the unwanted character removed in the preprocessing phase?
6. The variable width and height of the text-line images have been fixed to 48 pixels by locking the aspect ratio. What would be the Reason for the fixed Width and Height?
7. The reason behind the selection of Evaluation metric i.e. Levenstein Edit Distance (LED) for this work. What was the reason for this LED selection?
8. What were the other confusion letters other than stated in the discussion part and the reason for the confusion letters.
9. As the result is check on any commercial/open-source OCR stated in the introduction part of the paper.

---

## Round 0.2 · accepted · Accept

The paper is now acceptable.

Reviewer 1 ·

Basic reporting

All the mentioned are improved in this version of the paper.

Experimental design

All the mentioned are improved in this version of the paper.

Validity of the findings

All the mentioned are improved in this version of the paper.

Additional comments

All the concerns are addressed, the paper are accepted. The author shall check once the paper for the typo and grammatical mistakes/error. The number & caption of Figure and Table may be check.

Reviewer 2 ·

Basic reporting

The manuscript are refine and concise for the readers to gain the main idea. However the following section may be look again for typos/misspell if any. The introduction section and Literature review are good enough, but if possible add the worth of OCR application in the introduction section in a line.

1- Please add the worth of OCR application in Introduction Section
2- Have a look at general for typos if any in the manuscript.

Experimental design

Na

Validity of the findings

Na

Additional comments

Na

Reviewer 3 ·

Basic reporting

No further comments

Experimental design

No further comments

Validity of the findings

No further comments

Additional comments

No further comments